# Exploring perceived walkability in one-way commercial streets: An application of 360˚ immersive videos

**Dao Chi Vo, Jeongseob Kim** *

Department of Civil, Urban, Earth and Environmental Engineering, Ulsan National Institute of Science and Technology, Ulsan, South Korea

* jskim14@unist.ac.kr

**Data Availability Statement:** All relevant data are within the manuscript and its Supporting information files.

**Funding:** This research was supported by the National Research Foundation of Korea (NRF: 2021R1A2C2011106). The funders had no role in

## Abstract

This study explores the perceived walkability of one-way commercial streets by utilizing immersive 360-degree virtual reality (VR) videos. While one-way roads are often introduced to facilitate smooth traffic flow on narrow roads, providing safe and walkable environments for pedestrians on the one-way roads is crucial, especially in commercial areas with heavy pedestrian traffic. We recruited 40 students to assess the perceived walkability of one-way roads based on ten VR scenarios. For each scenario, variables such as road width, one-way traffic status, vehicle approach direction, and the presence of sidewalks were configured differently. This study found that while there is awareness of one-way road types, the presence of sidewalks is considered critical factor contributing to enhanced perceived walkability on narrow commercial street. In the condition of narrow road width, one-way roads are the only applicable road layout to have a sidewalk, implying the potential of one-way roads for better walkability. Besides, the study also revealed the high correlation between five dimensions of perceived walkability, emphasizing their role to enhance perceived walkability in a setting of one-way roads. This study's findings could be utilized for more systematic walkability assessments and urban design improvements, especially in constrained road spaces.

## Introduction

Since the postmodern era, transforming car-centric environments into pedestrian-friendly ones, where people feel safe and comfortable, has been considered a core planning direction for equitable, accessible, livable, and sustainable communities [1, 2]. Among transportation planning practices aimed at creating pedestrian friendly street environments, one-way road systems have long been a very controversial topic, particularly in commercial areas where heavy vehicle and pedestrian traffic coexist. There has been ongoing debate over the walkability of one-way roads. A group of researchers argued that although one-way roads are advantageous for facilitating vehicle traffic with a larger capacity for automobiles [3], pedestrians and cyclists tend to have more collision risk and less accessibility to their desired destinations due

study design, data collection and analysis, decision to publish, or preparation of the manuscript.

**Competing interests:** The authors have declared that no competing interests exist.

to the increased speed of driver's tendencies and the rerouting of long road [4–7]. However, other researchers argue that one-way road systems reduce traffic confusion, contributing to the safety of both vehicles and pedestrians, especially for the elderly, who have diminished eye movement, delayed perception, and slow reaction times [8–10].

In spite of the debate on the walkability of one-way roads, research on pedestrian environments on one-way roads remains relatively limited because many walkability studies have focused on conventional two-way road environments. Only a few studies have been conducted on how pedestrians on one-way roads perceive the safety and ease of walking across diverse urban forms [9, 11]. These studies have been conducted on the functional performance of one-way roads, such as traffic capacity and speed, but walkability has rarely been explored. Therefore, this study aims to explore the multifaced perceived walkability of one-way commercial streets, addressing the existing knowledge gap regarding walkability on one-way road systems. Specifically, this study addresses three research questions: (1) How different are various dimensions of perceived walkability across typical types of narrow commercial streets? (2) Do pedestrian positions that allow walking in or against traffic flows affect perceived walkability of one-way roads? (3) What factors contribute to the perceived walkability of one-way roads?

To answer these research questions, this study conducts Virtual Reality (VR) experiments that allow pedestrians to evaluate the walkability of various street environments, and analyzes the evaluation results using various statistical techniques. Further, the study examines the intricate cognitive processes involved in walkability across one-way roads, including perceived walkability in back-to-traffic/face-to-traffic positions. The experiments using immersive, 360-degree VR videos focus on assessing various dimensions of perceived walkability, such as safety and convenience, in commercial streets with one-way roads. VR is an effective tool for providing participants with immersive environments, allowing them to evaluate varied street environments. Previous research has shown that immersion creates a significant sense of being present and generates a virtual environment that closely mimics the physical one [12, 13]. It permits more precise assessments of subjective and objective cognitive responses in specific settings, such as streetscapes or landscapes [14]. Employing an immersive experimental environment can enhance our understanding of how street design influences subjective walkability.

The findings of this study based on the VR experiments could contribute to the design of pedestrian-friendly streets and enhance our understanding of perceived walkability in one-way road systems. This study focuses on typical commercial streets in Korean cities and is based on the walkability evaluations from a limited number of participants, which restricts the generalizability of the findings to other countries. However, it is significant as a pioneering study that comprehensively evaluates the perceived walkability of one-way commercial streets using VR and derives urban planning and design implications for improving street environments from the perspective of pedestrians.

## Literature review

A growing body of research focused on how to evaluate pedestrian-friendly environments or walkability and how to improve them has emerged. Studies have been conducted to measure walkability based on built environment factors such as land-use mix, density, public transit accessibility, and proximity to various services. In line with this approach, metrics such as the Walk Score have been developed and widely utilized to evaluate walkability. In addition to efforts to quantify built environments, researchers have conducted studies on individuals' subjective evaluations of pedestrian-friendly environments, known as perceived walkability. Unlike objective walkability, which focuses on the physical characteristics of urban design,

**Table 1. Different dimensions of perceived walkability in walkable studies.**

|  | Alfonzo (2005) [17] | Cerin et al. (2007) [23] | Jiang et al. (2012) [20] | Jun et al. (2015) [24] | Sun et al. (2016) [21] | Gan et al. (2021) [19] | De Vos et al. (2022) [16] | Basu (2022) [18] | Liao et al (2022) [22] |
|---|---|---|---|---|---|---|---|---|---|
| Feasibility of walking | X |  |  |  |  |  | X |  |  |
| Accessibility | X | X |  |  |  |  |  |  |  |
| Ability to walk to store/shop |  |  |  | X |  |  |  |  |  |
| Safety from traffic | X | X | X |  |  | X |  | X |  |
| Safety from crime | X | X |  |  |  |  |  | X |  |
| Ease of walking |  |  |  | X |  |  |  |  |  |
| Difficulty of crossing the street |  |  |  |  | X |  |  |  |  |
| Convenience for walking |  |  |  |  |  |  | X |  |  |
| Comfort of walking | X |  | X |  |  |  | X |  | X |
| Enjoyment of walking |  |  | X |  |  | X |  |  |  |
| Pleasantness of walking | X |  |  |  |  |  | X | X |  |
| Friendliness |  |  |  |  |  |  |  | X |  |
| Attractiveness |  |  |  |  | X |  |  | X |  |
| Happiness |  |  |  |  |  |  |  |  | X |
| Annoyance |  |  |  |  |  |  |  |  | X |
| Satisfaction while walking |  | X |  |  |  |  |  |  | X |
| Stimulation or willingness to walk |  |  |  |  |  |  | X |  |  |

perceived walkability represents people's diverse cognitive responses possibly shaped by their experiences, preferences, abilities, and cultural context [15, 16]. It could provide a more comprehensive understanding of individuals' walking behaviors in a given street design.

Research on subjective walkability has continued to evolve since Alfonzo (2005) [17] proposed five dimensions of walkability based on a synthesis of existing studies, as summarized in Table 1. Alfonzo (2005) [17] suggests five dimensions of perceived walkability based on walking needs: feasibility, accessibility, safety, comfort, and pleasurability. Feasibility is a key indicator of walking ability (considering factors such as age and physical condition), while accessibility assesses how a particular area facilitates street access. Safety refers to how secure individuals feel when walking, particularly concerning both traffic and criminal risks. In particular, safety from traffic is fundamental in assessing walkability, as emphasized in numerous transport and urban studies that focus on traffic as a deterrent to walking [17–20]. Comfort relates to the ease and convenience of walking activities. The dimensions of comfort and convenience are often used interchangeably to reflect the physical conditions of streets and urban environments, ensuring that walking is practical, comfortable, and easy [17, 19, 21, 22]. Pleasurability concerns the aesthetic appeal of the environment that enhances the enjoyment of walking. Recent studies have expanded the dimension of pleasurability to include aspects such as attractiveness, friendliness, and emotional factors like happiness and satisfaction [18, 21, 22]. In addition to these five dimensions, De Vos et al. (2022) introduced walking stimulation as a measure to further capture perceived walkability [16]. Evaluating perceived walkability across these dimensions contributes to a holistic understanding of the walking environment, considering both functional and psychological aspects.

As mentioned earlier, the perceived walkability of one-way roads is a controversial topic, which is the main focus of this study. Researchers argue that one-way roads improve vehicle traffic capacity but may increase collision risks and reduce accessibility for pedestrians and cyclists [5, 7], while others claim these systems reduce traffic confusion and enhance safety, especially for the elderly [8–10]. In some cases, particularly in commercial or mixed-use contexts, one-way roads can provide pedestrian-scaled town centers, as proposed by Peter Calthorpe, targeting increased density of road networks with narrow, interconnected streets, contributing to reduced traffic flow, and promoting walking and cycling [25, 26]. In this way, the places with one-way roads could provide higher levels of happiness and "livability" than those with two-way roads with similar traffic volume [27]. Some researchers have specifically focused on the effect of vehicle traffic direction in one-way roads. Facing traffic provides pedestrians with visual information about automobiles, thereby making them safer [28]. In this situation, a pedestrian may sense a lower risk of being hit by a vehicle, promoting walkability. Therefore, the benefits of one-way roads are context-sensitive; a nuanced exploration of pedestrian-oriented road design rather than a categorical judgment of its comparison with two-way roads, as some researchers suggest [29, 30].

## Materials and methods

### Ethics information

The Institutional Review Board (IRB) of the Ulsan National Institute of Science and Technology (UNIST) granted approval for this study (Protocol No. UNISTIRB-23-002-A), including the experimental protocol, to safeguard the rights and well-being of participants under the project titled "Exploration of multisensory perception in commercial street: the virtual reality approach". Based on this approval, the experiment was implemented in accordance with applicable guidelines and regulations to ensure compliance with ethical standards. The experiment was carried out at the Urban Planning & Analytics lab, UNIST. The laboratory took charge of participant recruitment and payment processing to ensure the security of both experimental data and personally identifiable information. Advertisements at UNIST in Ulsan, South Korea, were utilized to recruit participants. To be considered for recruitment, participants should be between the ages of 18 and 39, have normal hearing, and have no prior history of brain surgery or mental illness. From July 8–20th, 2023, we recruited and selected a total of 40 undergraduate and graduate students at UNIST, with a mean age of 23.9 and an age range of 19–30. On the experiment date, we informed the selected participants and required them to complete a written informed consent form in accordance with the IRB. We completed the trial in less than forty-five minutes and compensated each participant with approximately $20 USD.

### Experimental setting: VR videos and scenarios

We selected three conventional one-way commercial streets, which have different layouts, among Ulsan's major commercial districts. For a comparative evaluation of the perceived walkability, a typical two-way commercial street and a pedestrian-only street were also selected. As shown in Table 2 all 360˚ videos were recorded during off-peak hours. The road width of the case streets was 8 meters, except for a wider one-way road (12 meters wide), which is included to examine the effect of a wider sidewalk on perceived walkability. The recording spot was located in the middle of the road segment to provide an ideal street environment for walking and crossing the midblock of a narrow one-way road. We filmed another commercial street, in addition to these five roads, to familiarize the participants with the VR environment.

**Table 2. Description of experimental street scenarios.**

| | ID | Place (Abbreviation) | Road types | Road width (meters) | Vehicle approach direction | # of parked cars | # of pedestrian |
|---|---|---|---|---|---|---|---|
| Pilot | S0 | Samsan-dong | Two way with sidewalk | 14 | Forward | 1 | 10 |
| One-way street | S1 | Seongnam-dong (sidewalk 8_forward) | One way with sidewalk | 8 | Forward | 1 | 4 |
| | S2 | Seongnam-dong (sidewalk 8_backward) | One way with sidewalk | 8 | Backward | 0 | 6 |
| | S3 | Seongnam-dong (sidewalk 12_forward) | Wider one way with sidewalk | 12 | Forward | 0 | 8 |
| | S4 | Seongnam-dong (sidewalk 12_backward) | Wider one way with sidewalk | 12 | Backward | 2 | 5 |
| | S5 | Ulsan university (shared 8_forward) | One way without sidewalk | 8 | Forward | 3 | 4 |
| | S6 | Ulsan university (shared 8_backward) | One way without sidewalk | 8 | Backward | 1 | 10 |
| Pedestrian | S7 | Seongnam-dong (pedestrian 8_forward) | Pedestrian only | 8 | Forward (No vehicle) | 0 | 17 |
| | S8 | Seongnam-dong (pedestrian 8_backward) | Pedestrian only | 8 | Backward (No vehicle) | 0 | 18 |
| Ref: two-way | S9 | Ulsan university (two way 8_forward) | Two way | 8 | Forward | 6 | 10 |
| | S10 | Ulsan university (two way 8_backward) | Two way | 8 | Backward | 2 | 6 |

The videos were filmed during the day on the whole week to ensure the relative balanced pedestrian and vehicle volumes could be captured across road types. On recording days, the environment needed to be sunny and the temperature 17–18°C, ideal for a walk. On site, the Insta 360 X3 recordings lasted for 20 minutes. The video resolution was 5760 × 2880 (5.7 K) with stereo sound. Similar to relevant studies, the camera was installed 1.5 meters above the ground to obtain eye-level videos [12, 31]. Next, the acoustic environment of the street settings was recorded simultaneously using Zoom H2N to produce an enhanced immersive environment. Through VR, Zoom H2N generates spatial audio, analogous to hearing in real life, and provides an immersive and dynamic experience. These recordings were then split into numerous one-minute videos with different pedestrian and automobile flows. The length of the clips was designed for 60 second segments, similar to those used by [2]. This length of time was deemed appropriate to prevent VR sickness and ensure equal viewing conditions. We synchronized 360-degree selected videos with spatial audio and then pre-processed them with Adobe Premium Pro 2023 to ensure high-quality videos for immersive environments.

As shown in Table 2 and Fig 1, the experiment consisted of 10 scenarios. We configured each of the five roads using scenarios in both directions. The reason for setting up two scenarios for each road was to analyze the effects of the vehicle approach direction on one-way roads. Additionally, there were slight differences in the built environment, such as building heights, and street furniture for each direction, allowing us to consider the impact of environmental differences within the same commercial street. Five additional scenarios, in which all sounds were muted for each road type scenario, were also added to test the effect of sound on perceived walkability. However, this study did not use the results of these five scenarios, as they were beyond its scope. Therefore, each participant evaluated the walkability of 16 scenario environments, including a basic scenario for adaptation purposes (Scenario 0).

HTC Vive Pro VR headsets projected each scenario in an immersive setting. The Vive Pro headset offers stereo vision with 2880 × 1660 pixels per eye and a 110-degree field of view.

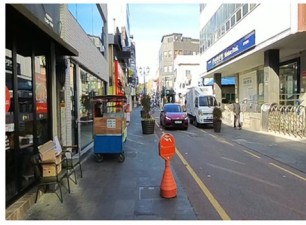

S1: Sidewalk 8_forward

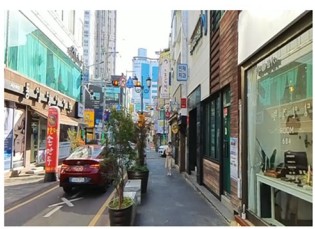

S2: Sidewalk 8_backward

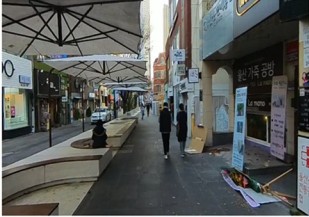

S3: Sidewalk 12_forward

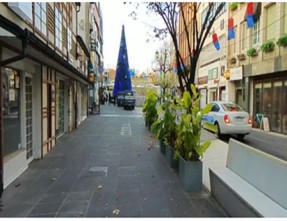

S4: Sidewalk 12_backward

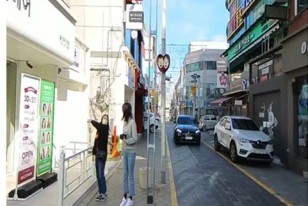

S5: Shared street_forward  S6: Shared street_backward  S7: Pedestrian_forward  S8: Pedestrian_backward

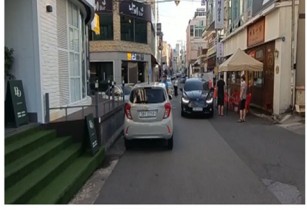
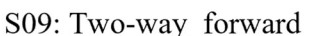

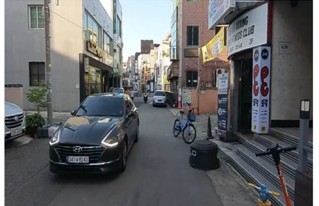

S09: Two-way_forward  S10: Two-way_backward

**Fig 1. The snapshot of experimental street scenarios.**

Spatialized environmental noise can be heard using headphones. Due to the participants' fixed positions, the wired configuration ensured VR performance stability during testing. Participants streamed the video via a Steam VR-compatible Virtual Desktop.

### Experiment procedure

Fig 2 shows a brief description of the procedure. Each participant was notified via email one day before the VR experiment's scheduled occurrence that alcohol consumption should be avoided to minimize health risks during the experiment. Before VR participation, each individual was required to provide informed consent and complete an initial questionnaire. The questionnaire inquired about personal information, such as demographics, health status, driving experience, collision and traffic accident history, habits of walking on commercial streets, familiarity with virtual environments, and other relevant details.

Next, we instructed the participants to sit in a designated seat and gave them instructions on how to wear the headset properly to reduce the likelihood of experiencing VR sickness or discomfort from the interface elements used in this research. The participants were instructed to watch in a fixed direction while able to freely rotate their headset, but not their entire body, allowing them to merge comfortably into the VR environment ahead of them and evaluate each direction of the roads. Subsequently, each participant was shown 16 one-minute videos, including the trial scenario, and instructed to complete a second questionnaire after exposure to each scenario. The initial trial scenario (Scenario 0) was excluded from the analysis because

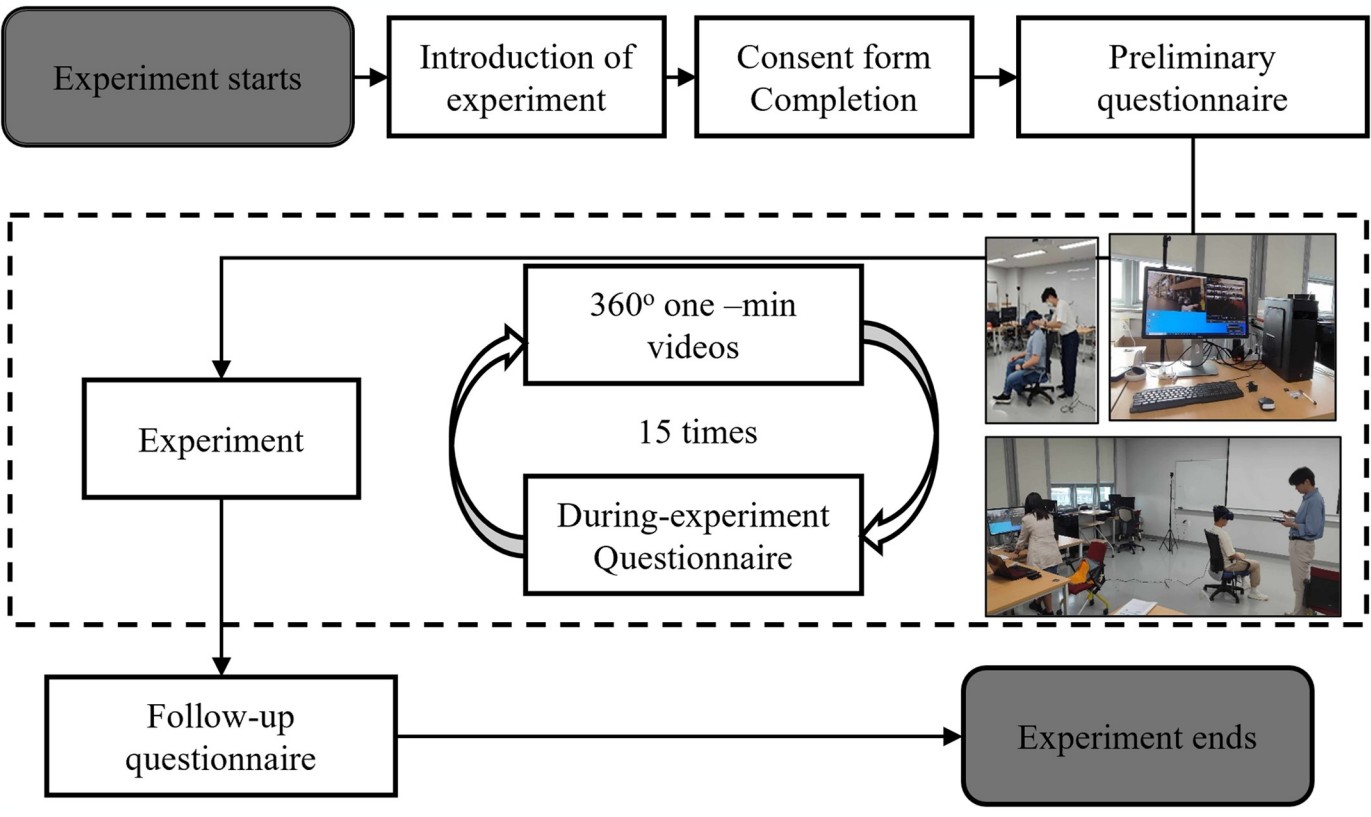

**Fig 2. An overview of the experimental procedure.**

of its limited purpose of familiarizing participants. A random assignment of the remaining scenarios was conducted to minimize the impact of cognitive memory bias. To prevent VR sickness, which could have resulted from continuous, repetitive VR exposure, every person received a brief, thirty-second break after the seventh trial. We also excluded the results of scenarios 11–15, which evaluated sound based on street experiences, from this study, as previously noted.

After the experimental phase, the subjects removed their headsets and were instructed to complete a follow-up questionnaire. The follow-up questionnaire included questions on scenario difference detection, virtual reality immersion, and sickness. To evaluate the important street elements in the experimented road layout, the participants were asked to identify three street elements affecting pedestrian traffic safety in this questionnaire. Following the successful completion of the research, which was catalyzed by this survey, the participants received the noted compensation.

### Measurement

Considering diverse comprehensive dimensions of perceived walkability as shown in Table 1, the study evaluated different street environments employing a VR approach, involving five specific dimensions of perceived walkability. It includes safety, convenience, attractiveness, and willingness to walk. Since healthy college students assess the given street environment, we consider the VR experiment to be well-met in terms of feasibility and accessibility; therefore, we excluded these aspects. Since safety on one-way roads is a controversial issue, we

introduced two sub-dimensions of safety for a more comprehensive assessment: safety for walking and safety for crossing. These distinctions allow for a nuanced exploration of safety in both linear (along the walking path) and horizontal (crossing mid-street) directions. Additionally, we measured the ease of walking using perceived convenience [16], and evaluated the aesthetic appeal of the street environment using perceived attractiveness [18, 21]. The study selected both terms, convenience and attractiveness, to ensure that Korean participants comprehend and accurately evaluate street settings according to their perceived walkability without language bias. In Korean, "convenience" and "attractiveness" are intuitively understandable and easier to evaluate compared to other similar walkability concepts that can be used interchangeably. We delivered both Korean and English versions of the survey simultaneously. We added the willingness to walk on a given road to establish a link between the subjective evaluation of walkability and the perceived walking intention, in line with relevant studies [16, 32, 33]. Finally, we measured the overall perceived walkability by calculating the mean value of the five aforementioned dimensions, of which four focus on the place qualities related to physical design and the willingness to walk, referring to planned behavior.

For five assessments, we employed a Likert scale ranging from 1 (lowest) to 5 (highest) in response to the question, 'How do you evaluate the real-time scene?' for each given scenario. For instance, we asked participants to self-rate specific dimensions, such as the attractiveness of the street environment, using this five-level scale after completing the VR experience (See S1 Questionnaire). This approach provided a consistent, structured method for evaluating across all scenarios, ensuring uniformity in participant responses.

## Statistical analysis

A total of 40 students participated in each of the 10 VR scenarios, resulting in 400 experimental responses collected. The Cronbach's alpha test was adopted in the study to assess the reliability of the questionnaire before statistical analysis. The scale considered acceptable ($\alpha = 0.95$) based on the results, indicating the reliability and acceptability of the survey results. The study employed analysis of variance (ANOVA) as a statistical technique to examine various aspects of perceived walkability by comparing mean differences across ten road scenarios. Furthermore, the study used Spearman's rank correlation matrix to examine the correlation among various measures of perceived walkability (See Fig 3). Additionally, this study utilized multilevel ordered logistic regression models to investigate the effects of road type and the direction of vehicle traffic on perceived walkability. The other variables, such as, variation of the building plane [16, 34], the presence of bench [35], a number of parked car and people [32, 33, 36], and personal characteristics (age, gender, the major transport mode, the experience of transport accident) [4, 17] were treated as controlling variables. The multilevel ordered logistic model allows for perceived walkability ratings as ordinal responses [37] to data while simultaneously addressing repeated assessments at the participant level, similar to previous relevant VR studies [38–40]. The multilevel model with the random intercept of each participant can minimize the bias of unobserved variables in individual subjects.

## Results

As shown in Table 3, the study recruited 40 undergraduate, graduate, and researchers from Ulsan National Institute of Science and Technology in Ulsan, Korea (62.5%, 30%, and 7.5%, respectively). Their average age is 23.9 years, with a range of 19 to 30 years (40% male, 60% female). Most participants (97.5%) were in good health during the VR experiment, and a large proportion (82.5%) were licensed drivers. However, commuting habits were a more relevant factor, possibly affecting walkability in this study, in which 75% of them commute to

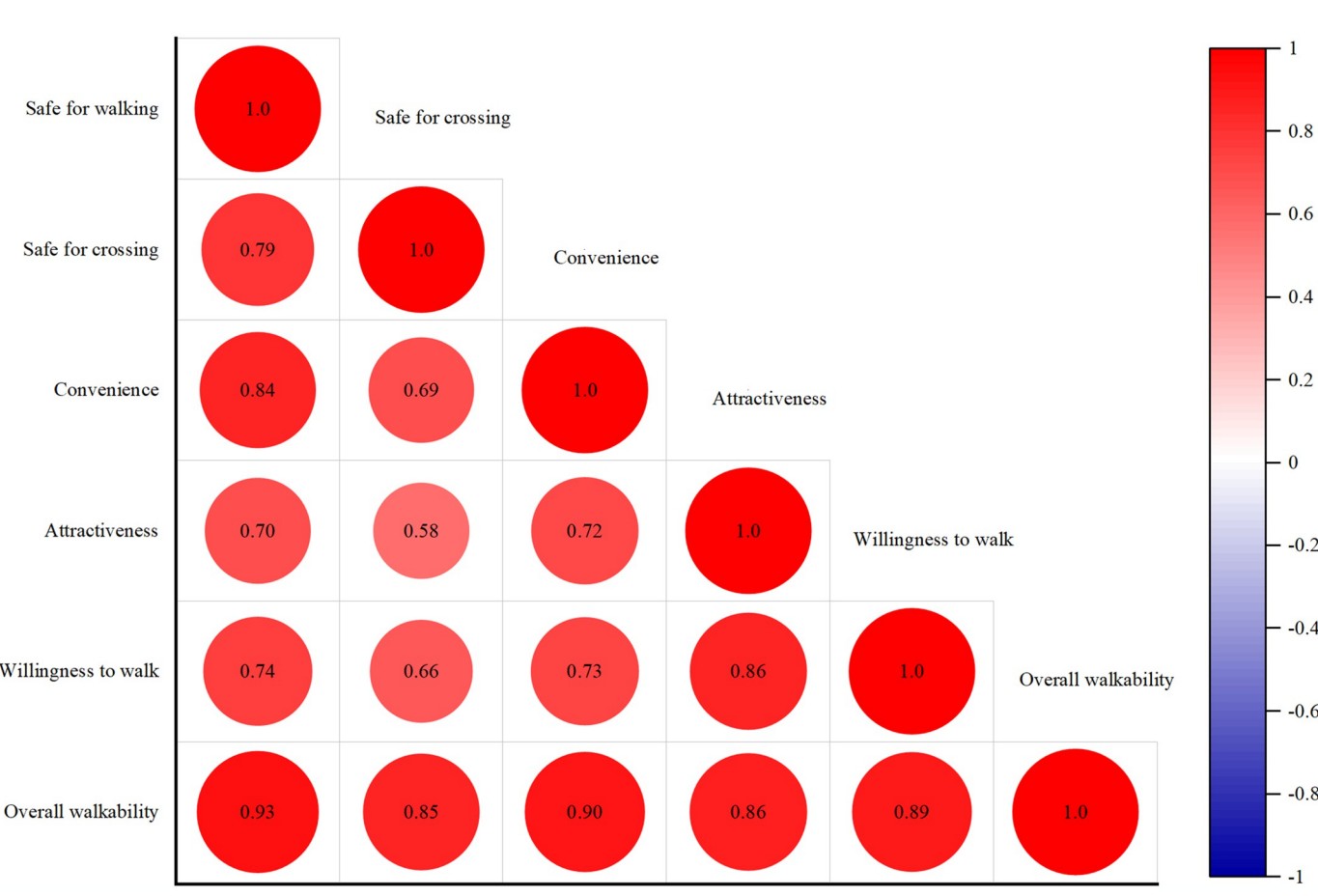

**Fig 3. Spearman's rank correlation matrix of five dimensions of perceived walkability.**

commercial streets by public transportation, 22.5% by automobile, and 2.5% on foot. Eight of them (20%) had been in a collision, with pedestrian-vehicle or vehicle-only accidents accounting for 50 percent each. The vast majority of them (72.5%) have prior familiarity with virtual reality (VR) technology. We treated the participants' sociodemographic characteristics (age, gender, commuting habits, and VR familiarity) as controlled variables to prevent bias in the study of perceived walkability. Based on the follow-up questionnaire, respondents rated scenario identification as 3.85, VR immersion performance as 4.3, VR sickness as 1.3, and eye fatigue at 2.1 on a 5-point Likert scale. Therefore, the entire procedure produced credible results for the subsequent analysis.

Regarding the relationship between the features of perceived walkability, as depicted in Fig 3, the study found the strongest correlation between safety for walking and convenience (0.84), and between attractiveness and the willingness to walk (0.86) among the five individual dimensions of perceived walkability. In contrast, safety for crossing exhibits a relatively low correlation with other individual dimensions of perceived walkability; however, the correlation coefficient remains high, ranging from 0.58 to 0.69. Overall perceived walkability demonstrates a high correlation of over 0.85 with individual variables, particularly being strongly correlated with safety for walking (0.93) and convenience (0.90).

**Table 3. Demographic information of participants.**

| Items | Value | Percentage (%) |
|---|---|---|
| **Participants (N)** | 40 | 100.0% |
| Undergraduate | 25 | 62.5% |
| Graduate | 12 | 30.0% |
| Researchers | 3 | 7.5% |
| **Mean of age (years old)** | 23.9 | |
| **Age range (years old)** | 19–30 | |
| **Gender** | | |
| Male | 16 | 40.0% |
| Female | 24 | 60.0% |
| **Health issue** | | |
| None | 39 | 97.5% |
| Yes | 1 | 2.5% |
| **Licensed Drivers** | | |
| None | 7 | 17.5% |
| Yes | 33 | 82.5% |
| **Frequent Travel mode** | | |
| automobile | 9 | 22.5% |
| by foot | 1 | 2.5% |
| public transport | 30 | 75.0% |
| **Accident History** | | |
| None | 32 | 80.0% |
| Yes | 8 | 20.0% |
| **Accident type** | | |
| Pedestrian-Vehicle (walking) | 4 | 50.0% |
| Vehicle-Only | 4 | 50.0% |
| **Familiarity with VR Technology** | | |
| None | 11 | 27.5% |
| Yes | 29 | 72.5% |

The robustly substantial association between attractiveness and willingness to walk highlights the critical significance of an attractive street environment in potentially promoting deliberate walking activity, which are more emphasized in recent walkability studies [18, 41]. Interestingly, the study found relatively weak correlation between safety for crossing and other dimensions of perceived walkability such as convenience, attractiveness, and willingness to walk. It could be because of the nature of the act of crossing highly involving risk-based behavior, which may not directly influence perceived walkable experience in the VR experiment [40, 42]. In the narrow commercial streets with a width of 8–12 meters studied in this research, crossings are relatively safer compared to wider streets, and since actual crossings mainly occur at intersections rather than mid-street, the importance of crossing safety may have been rated lower in the VR experiments. Therefore, although prior studies have indicated the critical importance of walking safety, we argue that safety alone is unlikely to be the most important factor; it may be context-dependent according to the act of crossing or walking. In a setting where crossings are inherently safe, other factors, such as convenience and attractiveness, should be considerably addressed to encourage people to walk. While safety is a fundamental dimension, our findings suggest that to create a successful walking environment calls for a holistic integration of safety, convenience, and attractiveness to shape overall perceived

walkability. The correlation-based finding could suggest priorities for designing effective interventions for urban walkability enhancement. Improvements in high correlation pairs may simultaneously elevate overall walkability, while low correlation ones require targeted interventions with more effort to ensure both dimensions reach optimal levels. As shown in Fig 4, we examined diverse aspects of perceived walkability in our research across different road scenarios. Panel (1) of Fig 4 showed the mean differences between forward and backward directions by road types regarding safety for walking and crossing. The differences were not statistically significant based on the Wilcoxon Signed-Rank Test, which was applied because the sample did not meet the normality assumption of the t-test. Since no statistical differences in perceived safety were found based on the direction of approaching vehicles, the comparison of perceived walkability across road types was conducted by aggregating the values from both directions. Panel (2) of Fig 4 displayed five dimensions of perceived walkability across road types, which were statistically significant at least at the 5% level of significance based on the ANOVA test.

One of the initial and main hypotheses in this study was that the vehicle approach directions influence the perceived walkability of one-way roads, particularly in terms of safety. This is because the visual and auditory perception of vehicles' approach (forward direction: the position enabling one to perceive the oncoming vehicle) may provide a better recognition of vehicle-related risk compared to the auditory perception of vehicles' approach (backward direction) on one-way roads. However, the results of the evaluation of the perceived safety in each direction did not show any significant differences on the one-way roads, as shown in Fig 4a. Similar to pedestrian-only and two-way roads, the perceived safety between forward and backward directions has no statistical differences based on the Wilcoxon Signed-Rank Test. These results imply that the vehicle approach direction has no effect on perceived safety in the cases of narrow commercial streets. However, these results could also be attributed to the relatively lower number of vehicles and less risky situations in the experimental scenarios.

Overall, narrow one-way roads have higher perceived walkability compared to that of two-way road. The participant rated ratings of perceived walkability according to the following ranking scheme: pedestrian road scored the highest level of perceived walkability scores, followed by wide sidewalks-one way road and narrow sidewalks-one way road with lower scores of perceived walkability. Following these was the shared one-way road, where there is no separated sidewalk. Lastly, two-way roads without a separate sidewalk had the lowest scores, indicating possible traffic confusion arising from vehicular traffic flows influencing the participants' perceived walkability [9, 43]. The findings demonstrated variations in perceived walkability ratings across different road designs, with statistical significance at a confidence level of at most 5%.

The different rating for each road type is derived from the presence of sidewalks, highlighting the critical role of sidewalks in improving walkability in the narrow road environment. Within roads with a constraint space for both pedestrians and vehicles, the addition of the sidewalk generated an approximately one-point increase in perceived walkability ratings (p< 0.001), compared with commercial two-way roads, where road spaces are known to be primarily dedicated to increasing the direction flows of vehicles. These walkable ratings' improvement was found highly on the aspects of safety for walking, convenience, and willingness to walk at approximately 1.4 points, while ratings for safety for crossing and attractiveness contribute to a rise by 0.5 and one point, respectively. Additionally, the results showed that the sidewalk design within narrow roads had higher walkable ratings than that of shared one-way roads, where the physical separation of vehicular traffic and pedestrians is removed. This improvement was particularly notable in the aspect of pedestrian convenience (an increase of 0.44 points), at a statistical significance of 0.01. It could underscore the pedestrian's preference

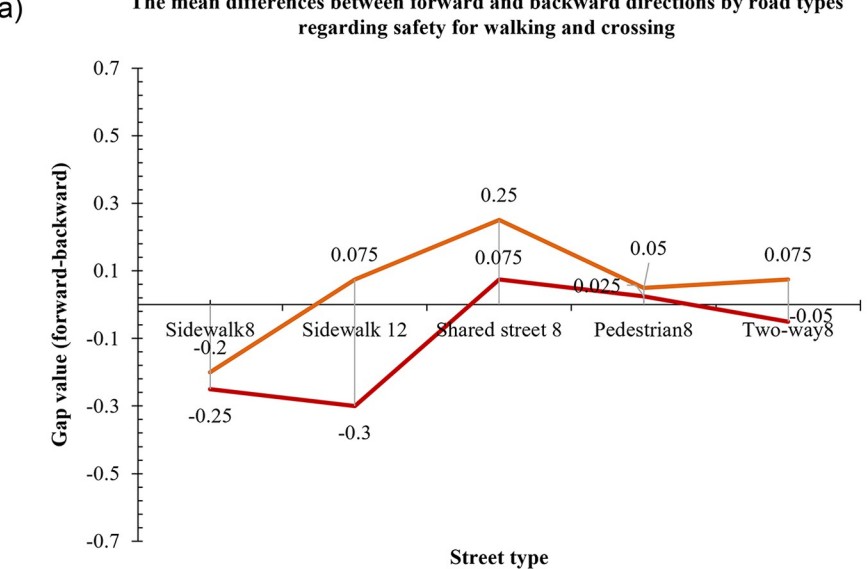

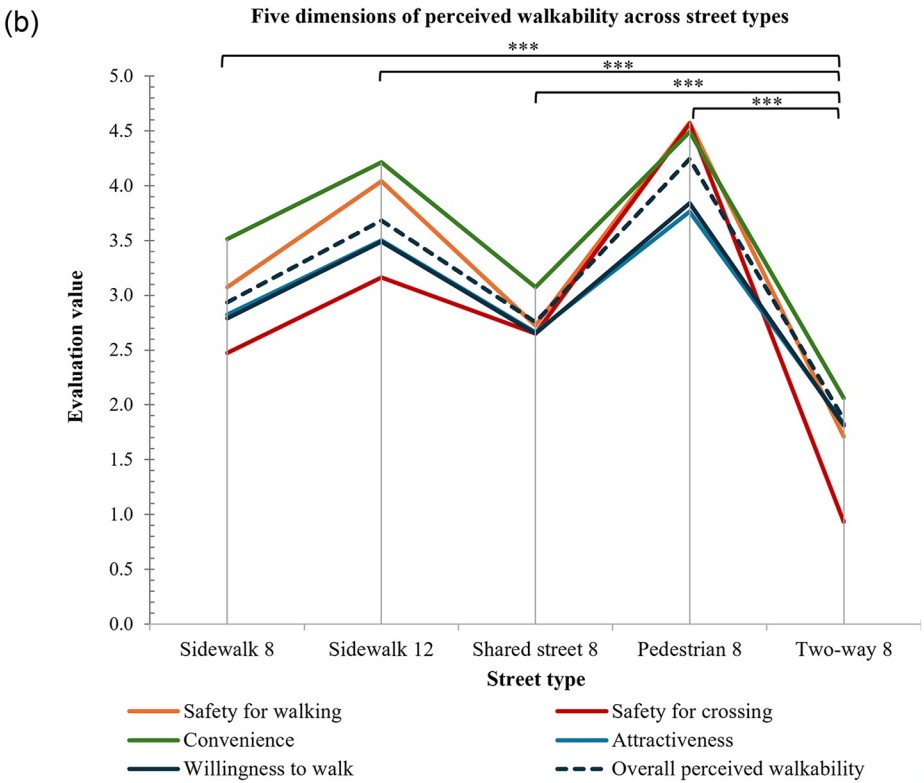

**Fig 4. Perceived walkability across road types.**

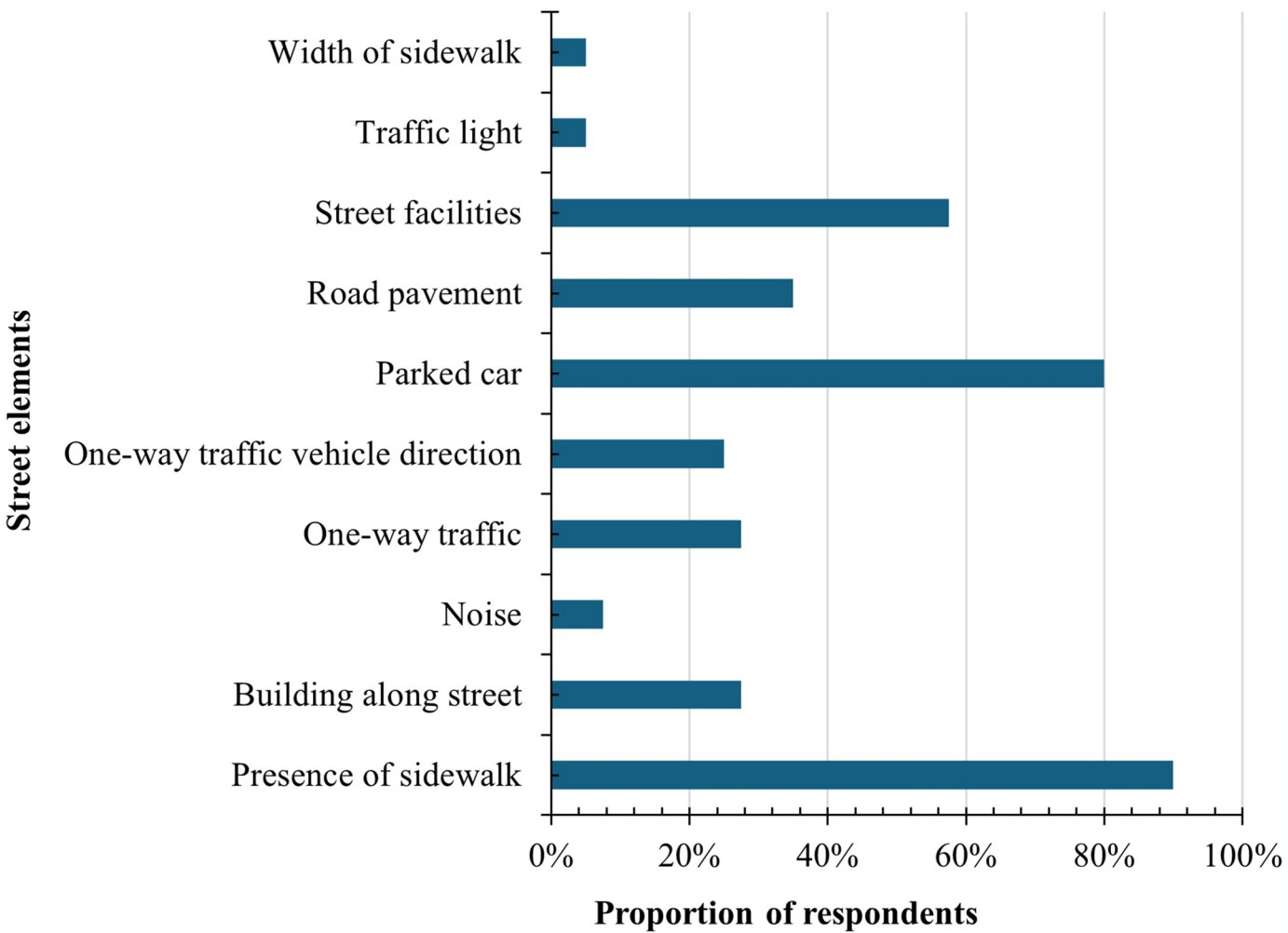

**Fig 5. The selected proportion of street environment factors believed to have influenced pedestrian traffic safety based on the scenarios of the VR experiment.**

for the value of maintaining visual separation between pedestrians and vehicles between road design layouts, consistent with relevant studies [44, 45]. The results of the post-experiment survey support the importance of sidewalk presence for safety and walkability. We asked participants to identify the most influential features of street elements on their perceived safety. Participants can choose up to three factors. The bar chart in Fig 5 presents the proportion of respondents selecting a particular choice as their determining factor. Notably, nearly 90% of respondents voted for the presence of sidewalks as the most significant factor. Besides these, the other elements: parked cars, street attributes, pavement, and buildings were less impactful, ranging from 35–80%. Following these are one-way roads and the directionality of one-way traffic accounted for perceived safety to varying lesser degrees (20–30%). These results imply that participants evaluate one-way roads' walkability higher not because of one-way itself but because of sidewalk presence.

However, only one-way roads could have a sidewalk in narrow commercial streets. Considering the width of each vehicle lanes, two way roads that are 8 meters wide cannot install sidewalks. At least in the setting of narrow commercial roads, the presence of sidewalks always

comes together with the one-way street layout. Therefore, one-way roads could be an effective system to improve perceived walkability in narrow street environments by providing a safe sidewalk.

The results also highlight the necessity of safe design for crossing on one-way roads. One-way roads with sidewalks showed the disparity between the safety for walking score and the safety for crossing score, while pedestrian-only roads and shared one-way roads didn't. These results imply that crossing is perceived as inherently riskier than walking in place, where the presence of interactions between pedestrians and vehicles can negatively affect perceived walkability. Pedestrians often cross streets in the middle of the streets rather than at the intersections, if crossing seems to be easier [46, 47]. This behavior could be more prevalent in narrow one-way commercial streets where pedestrians only need to cross one-vehicle lane and have more possibilities to cross to enjoy retail and commercial services on both sides of commercial streets [48]. Therefore, it is required to have greater consideration of crossing safety for better walkable narrow commercial road design to ensure that sidewalks fully benefit all aspects of pedestrian movement; for example, the appropriate street elements, such as benches and trees, should all be functionally designed to assist pedestrians in crossing safely.

The estimated results of the multilevel ordinal logistic model provide more evidence regarding the effect of road type and vehicle approach directions on perceived walkability in narrow commercial streets after controlling for other factors. The results of the intra-class correlation coefficient (ICC) ranged from 21% to 31%, implying the multi-level approach is acceptable.

As shown in Table 4, the results of the effect of road types present that one-way roads have higher perceived walkability compared to the two-way reference, consistent with the results of the ANOVA analysis above. In line with expectations, pedestrian-only streets had the highest odds ratios across all six dimensions of perceived walkability with statistically significant results (p < 0.001). In general, wide sidewalks, followed by narrow sidewalks and shared streets, showed progressively lower odds ratios, however, contributing positively to walkability. The overall perceived walkability showed a similar pattern to the individual measures of perceived walkability. Interestingly, the study did not find any statistically significant effects of road types on the safety for crossing in the narrow street between the two-way road (reference) and one-way roads. It suggests that while pedestrians evidently perceived safety when walking along the pathway, this does not necessarily extend to the perception of safety when crossing the street. It could be attributed to the absence of tangible and visual infrastructure for crossings, such as marked crosswalks or pedestrian signals [44, 45]. Without these, pedestrians may feel uncertain of their right-of-way when crossing in the constrained street, particularly in narrow streets where vehicles are in close proximity. Despite the potential for low vehicle speeds or volume on narrow streets, the absence of designated crossing areas could lead to uncertainty, thereby minimizing pedestrians' sense of safety when crossing.

The results also support the importance of sidewalks on one-way roads. For all five dimensions of perceived walkability, the one-way road with sidewalks showed an additional improvement in perceived walkability compared to those of the shared one-way road, and the positive effects are even higher in the one-way road with wider sidewalk.

The regression model also confirmed that there is no effect of vehicle approach direction on the perceived walkability. Although this effect was not statistically significant, contrary to our initial hypothesis, it does not imply an absence of risk detection during the walking process. Walking could be the result of interplay of complex cognitive and perceptual processes, which are affected by how pedestrians perceive walking situations in an environment. In particular situations, "walking without awareness" can occur as a highly automated habit, that allows a person to avoid obstacles with little to no awareness of the presence of obstacles in the

**Table 4. Result of multilevel (ordered logistic) regression models demonstrating the effect of road types and pedestrian's direction on perceived walkability.**

| term | Safety for walking | | | Safety for crossing | | | Convenience | | | Attractiveness | | | Willingness to walk | | | Overall perceived walkability | |
|---|---|---|---|---|---|---|---|---|---|---|---|---|---|---|---|---|---|
| | β | odds ratio | p | β | odds ratio | p | β | odds ratio | p | β | odds ratio | p | β | odds ratio | p | β | p |
| **Road type: (ref: two-way)** | | | | | | | | | | | | | | | | | |
| Sidewalk8 | 4.08 | 59.3 | 0.011 | 1.43 | 4.18 | 0.377 | 4.57 | 96.7 | 0.003 | 2.77 | 16.0 | 0.048 | 3.18 | 24.0 | 0.025 | 1.17 | 0.007 |
| Sidewalk12 | 6.03 | 417.0 | 0.000 | 3.04 | 20.9 | 0.040 | 6.39 | 593.4 | 0.000 | 4.20 | 66.4 | 0.001 | 4.77 | 118.4 | 0.000 | 1.82 | 0.000 |
| Shared8 | 3.37 | 29.1 | 0.002 | 2.14 | 8.5 | 0.052 | 3.31 | 27.3 | 0.002 | 2.64 | 14.0 | 0.007 | 2.97 | 19.4 | 0.003 | 1.04 | 0.001 |
| Pedestrian8 | 8.51 | 4951.6 | 0.000 | 7.69 | 2190.1 | 0.000 | 7.51 | 1822.0 | 0.000 | 6.28 | 531.8 | 0.000 | 7.28 | 1449.0 | 0.000 | 2.85 | 0.000 |
| **Pedestrian traffic view(ref: rear view)** | | | | | | | | | | | | | | | | | |
| Frontal view | 0.07 | 1.1 | 0.853 | 0.07 | 1.1 | 0.859 | -0.12 | 0.9 | 0.752 | -0.21 | 0.8 | 0.516 | -0.24 | 0.8 | 0.475 | -0.02 | 0.838 |
| **Control variables:** | | | | | | | | | | | | | | | | | |
| The presence of street elements (ref: No) | | | | | | | | | | | | | | | | | |
| Variation in building plane | 0.30 | 1.4 | 0.506 | -0.16 | 0.9 | 0.735 | 0.38 | 1.5 | 0.389 | 0.04 | 1.0 | 0.923 | 0.24 | 1.3 | 0.561 | 0.05 | 0.673 |
| The presence of bench | 0.64 | 1.9 | 0.204 | 0.39 | 1.5 | 0.447 | 0.36 | 1.4 | 0.487 | 0.64 | 1.9 | 0.183 | 0.56 | 1.8 | 0.258 | 0.22 | 0.135 |
| Street Occupancy | | | | | | | | | | | | | | | | | |
| No. people | -0.06 | 0.9 | 0.880 | -0.12 | 0.9 | 0.745 | 0.02 | 1.0 | 0.947 | -0.39 | 0.7 | 0.246 | -0.54 | 0.6 | 0.114 | -0.11 | 0.302 |
| No. parked car | 0.41 | 1.5 | 0.434 | 0.18 | 1.2 | 0.738 | 0.38 | 1.5 | 0.460 | 0.34 | 1.4 | 0.482 | 0.41 | 1.5 | 0.391 | 0.11 | 0.451 |
| Principle travel mode (ref: car) | | | | | | | | | | | | | | | | | |
| Public transportation | 1.88 | 6.5 | 0.001 | 2.34 | 10.4 | 0.001 | 1.96 | 7.1 | 0.002 | 1.14 | 3.1 | 0.028 | 1.33 | 3.8 | 0.013 | 0.65 | 0.003 |
| On foot | 4.11 | 60.7 | 0.005 | 4.91 | 135.7 | 0.011 | 4.60 | 99.6 | 0.007 | 3.96 | 52.2 | 0.005 | 4.59 | 98.4 | 0.002 | 1.66 | 0.004 |
| Gender (ref: female) | | | | | | | | | | | | | | | | | |
| Male | 0.32 | 1.4 | 0.410 | 0.73 | 2.1 | 0.162 | -0.03 | 1.0 | 0.953 | 0.11 | 1.1 | 0.762 | 0.49 | 1.6 | 0.200 | 0.12 | 0.433 |
| Age (years old) | 0.29 | 1.3 | 0.001 | 0.35 | 1.4 | 0.003 | 0.29 | 1.3 | 0.004 | 0.11 | 1.1 | 0.164 | 0.14 | 1.1 | 0.110 | 0.09 | 0.01 |
| Accident experience (ref: No) | | | | | | | | | | | | | | | | | |
| Yes | 0.08 | 1.1 | 0.857 | 0.40 | 1.5 | 0.516 | 0.05 | 1.0 | 0.934 | -0.35 | 0.7 | 0.440 | 0.08 | 1.1 | 0.870 | 0.02 | 0.929 |
| **ICC** | 0.23 | | | 0.38 | | | 0.31 | | | 0.21 | | | 0.23 | | | 0.32 | |

path of walking [49]. In this way, it may not be present in the investigation in the state of perception, but rather in the state of action with various degrees of stimulus. Therefore, further investigation of pedestrian actions at the levels of traffic flows on different street designs could significantly provide more dedicated knowledge of walkable cognition and behaviors.

Interestingly, regarding control variables, this study found that age significantly influences several dimensions of perceived walkability, despite conducting experiments exclusively with young adults aged 19 to 30. Older young adults are more likely to give higher ratings for safety for walking, safety for crossing, and convenience compared to their younger counterparts. Attractiveness and willingness to walk are less age-dependent, showing no statistically significant effects. This may be attributed to older pedestrians being more cautious and preferring more pedestrian-friendly environments, consistent with existing studies [16, 50].

## Discussion

This study employed a 360° immersive VR application to examine perceived walkability on five narrow commercial streets, with an emphasis on the one-way road type. The advanced method allows us to collect participants' responses through real-time experiences in a realistic street environment. It can overcome the problem of subjective recall from field trips and

questionnaires [41, 51], or static spatial data of GIS studies [52, 53], which cannot capture the full complexity of human cognitive evaluation. Specifically, the immersive VR experiment utilized in the study accurately replicated real-world street environments, providing richer experiences than static images or maps. This method allowed participants to observe and directly evaluate walkability components at the microscale. A fixed exposure time for each scenario in the VR experiment can consistently capture participants' immediate responses. This approach enables less biased and fairer comparisons across different street settings by providing a standardized exposure for street audits and mitigating biases associated with the varied durations of real-world pedestrian experiences [12]. Based on these advantages of the immersive VR experiment, this study employs a subjective walkability assessment for narrow commercial streets with various road layouts, focusing specifically on one-way roads. Therefore, the results and findings of this study using the VR experiment can broaden our understanding of the benefits that one-way roads can offer for a pedestrian-friendly commercial environment.

The finding of this study showed the variation of perceived walkability across road types, following the ranking scheme "pedestrian> wide sidewalk>narrow sidewalk> share street> two-way street". It suggests that one-way roads could be relatively advantageous in places where road space is limited for pedestrians and vehicles, requiring a comprehensive concern for the trade-off between pedestrian and vehicular flows. It is in line with some studies [9, 26] arguing that these designs enable a more simplified traffic pattern, and reduce traffic conflict points on the street, resulting in better walkability [8, 9].

This study highlights the importance of sidewalk installation in the setting of constrained street space. This is evident in the significant improvement in perceived walkability ratings on one-way roads with sidewalks, compared to two-way roads through ANOVA analysis and regression models. Narrow commercial streets without sidewalks could have a higher risk of pedestrian-vehicle accidents. Specifically, pedestrians tend to struggle with stationary obstructions and vehicular flow [9, 54] and they even can jaywalk in some circumstances. An automobile may arrive out of somewhere from another narrow street [55] and drivers may speed up in places without determining pedestrian and vehicular zones. However, the addition of a sidewalk on the narrow street combined with a one-way traffic system exhibited a significantly greater contribution to perceived walkability, especially in the aspect of safety for walking, convenience, and willingness to walk. Some studies have argued that pedestrians have a high degree of cognition of the tangible and visual design of street layout [44], which leads to their preference for pedestrian dedicated facilities [56]. Therefore, the presence of sidewalks on the narrow street make pedestrians easily receive and process the visual traffic information for their walking effectively. In this way, one-way roads with sidewalk could be an effective street design option because one-way roads are able to provide sidewalk on a narrow commercial street setting considering the required width of vehicle lanes. To sum up, while there is awareness relating to one-way road types, the addition of sidewalks emerges as the greater factor contributing to enhanced perceived walkability on the narrow commercial street. This regard leads to the proper recognition of the advantage of a one-way road lying in the ability to install a sidewalk on the narrow commercial street. By securing sidewalks, perceived walkability can be enhanced.

While this study focuses on the walkability assessment of one-way roads in Korean commercial areas, the results and findings may have implications that extend to similar urban settings in other countries. Commercial districts in cities worldwide face the dual challenge of addressing traffic congestion and pedestrian safety issues [25, 57]. To tackle these challenges, many cities across Asia, Europe, and North America are adopting one-way street systems within commercial districts, aiming to enhance traffic efficiency while creating pedestrian-

friendly street environments [25, 58, 59]. Despite differences in urban design, regulatory frameworks and travel behaviors, the underlying factors and mechanisms influencing the pedestrian experience and walkability can be similarly shared [41, 60–63]. Therefore, the study's findings, which focus on narrow one-way commercial streets in South Korea, could be transferable to diverse international contexts, enhancing the understanding of systematic walkability assessments based on user experience.

## Conclusion

With an emphasis on the layout of one-way roads, this study explores various dimensions of perceived walkability in the setting of typical narrow commercial streets in South Korea. As a unique methodological approach, the study used VR technology to create immersive environments for evaluating several scenarios of commercial streets. With the immersive environments experiment delivered to pedestrians, information regarding opinions and evaluations of one-way road scenarios were gathered for the following analysis.

Overall, this study suggests that one-way roads with a separate and wider sidewalk could improve perceived walkability compared to the shared one-way streets or the two-way streets. The one-way roads could simplify traffic and reduce conflict points, resulting in improved safety and convenience for pedestrians. Above all, the results of this study emphasize the importance of sidewalks in improving perceived walkability especially in constrained commercial street environments. This is because the presence of visible and unambiguous pedestrian signals plays a vital role in ensuring that pedestrians understand their right of way while walking. In sum, one-way roads that accommodate sidewalks and simplify traffic flow can enhance the perception of walkability, contributing to the establishment of a pedestrian-friendly environment which require an attentive incorporation of safety, convenience, and attractiveness to encourage walking habits.

However, this study has some limitations which can be addressed in future studies. This study included a small cohort of undergraduate and graduate university students aged 19–30 years. Moreover, the sample size was limited to 40 participants, generating 400 responses. This makes it challenging to generalize and extend the results across various age groups. Diversifying the subject cohorts by personal characteristics such as age, vision impairment, disability, driver, and cycling could improve perceptual knowledge, which is the basis for constructing a more inclusive walking environment. Because our experimental settings were recorded in real-world urban contexts rather than in virtual simulation places for VR investigations, we struggled to create fully controlled tests. Various traffic settings with different numbers of cars and pedestrians and diverse built environments with more case streets could provide more generalizable findings regarding the functions and effects of one-way roads. Additionally, the study reveals a high correlation between several components of perceived walkability, such as attractiveness and willingness to walk, suggesting a potential issue of collinearity among walkability variables. This interdependence among components necessitates cautious interpretation and highlights the need for future research to address collinearity concerns. In practice, recognizing this limitation is crucial for urban planners and designers to fully understand the impact of each walkability component on specific interventions, thereby contributing to the development of a more robust and reliable walkability evaluation framework.

## Supporting information

**S1 Questionnaire.**
(DOCX)

**S1 Data.**
(XLS)

# Acknowledgments

We would like to express our gratitude to the UNIST students who participated in the VR experiment.

# Author Contributions

**Conceptualization:** Dao Chi Vo, Jeongseob Kim.

**Data curation:** Dao Chi Vo.

**Formal analysis:** Dao Chi Vo.

**Funding acquisition:** Jeongseob Kim.

**Methodology:** Dao Chi Vo, Jeongseob Kim.

**Supervision:** Jeongseob Kim.

**Visualization:** Dao Chi Vo.

**Writing – original draft:** Dao Chi Vo.

**Writing – review & editing:** Jeongseob Kim.

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
