## [Decision Letter · Decision Letter 0]

3 Sep 2024

PONE-D-24-28229Exploring perceived walkability in one-way commercial streets: An application of 360° immersive videosPLOS ONE

Dear Dr. Kim,

Thank you for submitting your manuscript to PLOS ONE. After careful consideration, we feel that it has merit but does not fully meet PLOS ONE’s publication criteria as it currently stands. Therefore, we invite you to submit a revised version of the manuscript that addresses the points raised during the review process.

Two reviewers have provided their comments on your manuscript, which can be found below. The significance of this study, especially in terms of using VR over other methods or data used in the past, should be further clarified, as one reviewer specifically mentioned. Also, the results section and discussions should be expanded to focus on other aspects as well, i.e., not only safety but, comfort, attractiveness, etc. as well. 

We look forward to receiving your revised manuscript.

Kind regards,

Charitha Dias

Academic Editor

PLOS ONE

 [This research was supported by the National Research Foundation of Korea (NRF: 2021R1A2C2011106).].  

Please include this amended Role of Funder statement in your cover le

Additional Editor Comments (if provided):

Reviewers' comments:

Reviewer's Responses to Questions

**Comments to the Author**

1. Is the manuscript technically sound, and do the data support the conclusions?

Reviewer #1: Yes

Reviewer #2: Yes

2. Has the statistical analysis been performed appropriately and rigorously? 

Reviewer #1: Yes

Reviewer #2: Yes

3. Have the authors made all data underlying the findings in their manuscript fully available?

Reviewer #1: No

Reviewer #2: Yes

4. Is the manuscript presented in an intelligible fashion and written in standard English?

Reviewer #1: Yes

Reviewer #2: No

5. Review Comments to the Author

Reviewer #1: 1. Add Separate section on objectives of study with clearly stated objectives along with limitations of the study.

2. Conclusion should be separated by data. No references in conclusion section.

3. Add labels to x and Y axis in the figures along with legends.

Reviewer #2: Manuscript PONE-D-24-28229 explores the perceived walkability of one-way commercial streets by utilizing immersive 360-degree virtual reality (VR) videos. Although there are many studies on walkability, this study is unique in its use of VR. As a result, the presence of sidewalks is considered critical factor contributing to enhanced perceived walkability on narrow commercial street. The findings of manuscript PONE-D-24-28229 could be utilized for more systematic walkability assessments and urban design improvements, especially in constrained road spaces.

The reviewer evaluated the manuscript as revision. That is because the reviewer has some issues with your manuscript.

Introduction

Ln 94-95

Employing an immersive experimental environment can enhance our understanding of how street design influences subjective walkability.

The question here concerns the originality of your research. Nevertheless, you have not answered this question in your Discussion section. Walkability has been studied in a variety of ways, including questionnaires, GIS, etc. In contrast, did the methods used in your VR have the same results? Were they different results?

Material and Methods

Ln 111

a mean age of 26.95 and 111 an age range of 21–39

You need to describe in detail the socio-demographics of the participants. In your study, the questionnaire asked about personal information, such as demographics, health status, driving experience, collision and traffic accident history, habits of walking on commercial streets, familiarity with virtual environments, and other relevant details. You must clearly state in your manuscript whether these sociodemographic characteristics influenced the results.

Ln 187-197

Subsection Measurement

The content discussed in this subsection pertains to the concept of walkability in your study. Therefore, the description here should be described as a theoretical framework within or after the introduction, not as a material and method. Furthermore, I am concerned that your concept of walkability relies on Alfonz (2005). You should explain the validity of the three dimensions you have set by referring to more studies on walkability.

Ln 199-200

Among the five walking needs described above, we employed safety, comfort, and pleasurability.

This terminology of comfort and pleasurability differs from that of Figure 3-1. Is the comfort same as convenience? Is the pleasurability same as attractiveness?

Ln 204-205

the willingness to walk on a given road is added as a proxy of overall perceived walkability, consistent with relevant studies[23,24].

I disagree with the idea of measuring walkability by willingness to walk alone. You should make a subsection of the theoretical framework to reinforce the logic.

Ln 205-207

For these assessments, we employed a Likert scale ranging from 1 (lowest) to 5 (highest) in response to the question, ‘How do you evaluate the real-time scene?’ for each given scenario.

The results in Figure 3 show that the scores for willingness to walk and attractiveness are mostly similar. I am aware of the problematic questions in your study. Please describe how you asked all the questions, including the safety and comfort questions.

Result

Ln 226-227

As shown in Fig 3, we examined five aspects of perceived walkability in our research across different road scenarios.

Wouldn't it be appropriate to analyze Figure 3 from Scenario 1 to Scenario 10, respectively? I cannot understand why the forward and backward are combined in each of the scenarios in Table 1.

Ln 227-229

Panel (1) of Fig 3 displayed five dimensions of perceived walkability across road types, which were statistically significant at least at the 5% level of significance based on the ANOVA test.

Figure 3 should show the significant difference marks between the groups that were significantly different.

Ln 259-260

The results of the post-experiment survey support the importance of sidewalk presence for safety and walkability.

The results in Figure 3 show that the scores for willingness to walk and attractiveness are mostly similar. Looking at these results, is walkability not more important for attractiveness than safety? The reasons why we should focus on safety are not fully explained.

Ln 310-323

For all five dimensions of perceived walkability --- more dedicated knowledge of walkable cognition and behaviors.

I think this is the main part of your results. You should be a little more specific, describing your scores, including odds ratios. Another important result is that age is significantly related in the control group.

Discussion

The point you are missing in your discussion is the relationship between the elements of walkability. Your study was analyzed in terms of safety, comfort, attractiveness, and willingness to walk. However, with respect to the results, you focus too much on safety. You must add a paragraph on the relationship between the elements, based on the theoretical frame of reference.

6. PLOS authors have the option to publish the peer review history of their article (what does this mean?). If published, this will include your full peer review and any attached files.

Reviewer #1: **Yes: **Tejwant Singh Brar

Reviewer #2: No

---

## [Author Response · Author response to Decision Letter 0]

8 Oct 2024

Please, refer to the attached document about response to reviewers.

---

## [Decision Letter · Decision Letter 1]

1 Nov 2024

PONE-D-24-28229R1Exploring perceived walkability in one-way commercial streets: An application of 360° immersive videosPLOS ONE

Dear Dr. Kim,

Thank you for submitting your manuscript to PLOS ONE. After careful consideration, we feel that it has merit but does not fully meet PLOS ONE’s publication criteria as it currently stands. Therefore, we invite you to submit a revised version of the manuscript that addresses the points raised during the review process.

Reviewers would like to see some more <wt-ignore source="wt-feature-result" uuid="d964dec8-d7b2-4c44-846b-e6c9d19f87ed">discussion</wt-ignore> on this point, especially, how to generalize the findings of this study to other countries. 

We look forward to receiving your revised manuscript.

Kind regards,

Charitha Dias

Academic Editor

PLOS ONE

Reviewers' comments:

Reviewer's Responses to Questions

**Comments to the Author**

1. If the authors have adequately addressed your comments raised in a previous round of review and you feel that this manuscript is now acceptable for publication, you may indicate that here to bypass the “Comments to the Author” section, enter your conflict of interest statement in the “Confidential to Editor” section, and submit your "Accept" recommendation.

Reviewer #2: All comments have been addressed

2. Is the manuscript technically sound, and do the data support the conclusions?

Reviewer #2: Yes

3. Has the statistical analysis been performed appropriately and rigorously? 

Reviewer #2: Yes

4. Have the authors made all data underlying the findings in their manuscript fully available?

Reviewer #2: Yes

5. Is the manuscript presented in an intelligible fashion and written in standard English?

Reviewer #2: Yes

6. Review Comments to the Author

Reviewer #2: Manuscript PONE-D-24-28229_R1 was appropriately revised in the first round. First of all, I would like to thank the authors for their sincere response. However, some parts of this manuscript have not been sufficiently revised. The points were as bellow:

1. Discussion Ln 519-523

“ This immersive experience provides a more direct and detailed evaluation of walkability at the microscale, focusing on aspects like safety, convenience, attractiveness, and willingness to walk in this study. It enables us to generate fine grained data with higher accuracy to enrich the nuanced insights for different subjective aspects of walkability associated with specific street elements.”

The authors need to be more specific in describing how the results in this study relate to this point. The current explanation is not based on evidence.

2. sociodemographic characteristics Ln 326-336

“The study recruited 40 undergraduate, graduate, and researchers 326 from Ulsan National Institute of Science and Technology in Ulsan, Korea (62.5%, 30%, and 7.5%, respectively). Their average age is 23.9 years, with a range of 19 to 30 years (40% male, 60% female). Most participants (97.5%) were in good health during the VR experiment, and a large proportion (82.5%) were licensed drivers. However, commuting habits were a more relevant factor, possibly affecting walkability in this study, in which 75% of them commute to commercial streets by public transportation, 22.5% by automobile, and 2.5% on foot. Eight of them (20%) had been in a collision, with pedestrian-vehicle or vehicle-only accidents accounting for 50 percent each. The vast majority of them (72.5%) have prior familiarity with virtual reality (VR) technology.”

The author needs to add a table showing the results. For example, it is currently unknown what the responses of the 2.5% of people who did not answer "good health" were and what percentage of them were each.

3. Research Limitation Ln 340-347

“Regarding the relationship between the features of perceived walkability, as depicted in Figure 3, the study found the strongest correlation between safety for walking and convenience (0.84), and between attractiveness and the willingness to walk (0.86) among the five individual dimensions of perceived walkability. In contrast, safety for crossing exhibits a relatively low correlation with other individual dimensions of perceived walkability; however, the correlation coefficient remains high, ranging from 0.58 to 0.69. Overall perceived walkability demonstrates a high correlation of over 0.85 with individual variables, particularly being strongly correlated with safety for walking (0.93) and convenience (0.90).”

A high correlation between the components of the index indicates the risk of collinearity. The risk of collinearity means that the components of the index are incomplete. This problem should be noted as a research limitation and added to the conclusion.

4. Discussion: Generalization to other countries

Your research is on one-way roads in commercial areas in South Korea. However, in my opinion, there are some problems that are common to streets in commercial districts in other countries. Plos One is an international journal. The author should generalize your findings to other countries.

7. PLOS authors have the option to publish the peer review history of their article (what does this mean?). If published, this will include your full peer review and any attached files.

Reviewer #2: No

---

## [Decision Letter · Decision Letter 2]

2 Dec 2024

Exploring perceived walkability in one-way commercial streets: An application of 360° immersive videos

PONE-D-24-28229R2

Dear Dr. Kim,

We’re pleased to inform you that your manuscript has been judged scientifically suitable for publication and will be formally accepted for publication once it meets all outstanding technical requirements.

Kind regards,

Charitha Dias

Academic Editor

PLOS ONE

Additional Editor Comments (optional):

Reviewers' comments:

Reviewer's Responses to Questions

**Comments to the Author**

1. If the authors have adequately addressed your comments raised in a previous round of review and you feel that this manuscript is now acceptable for publication, you may indicate that here to bypass the “Comments to the Author” section, enter your conflict of interest statement in the “Confidential to Editor” section, and submit your "Accept" recommendation.

Reviewer #2: All comments have been addressed

2. Is the manuscript technically sound, and do the data support the conclusions?

Reviewer #2: Yes

3. Has the statistical analysis been performed appropriately and rigorously? 

Reviewer #2: Yes

4. Have the authors made all data underlying the findings in their manuscript fully available?

Reviewer #2: Yes

5. Is the manuscript presented in an intelligible fashion and written in standard English?

Reviewer #2: Yes

6. Review Comments to the Author

Reviewer #2: The authors have responded to all comments that reviewers. Finally, although Figure 1 shows photos of each street, I recommend also providing a map showing the relative locations of these streets.

7. PLOS authors have the option to publish the peer review history of their article (what does this mean?). If published, this will include your full peer review and any attached files.

Reviewer #2: No

---

## [Editor Report · Acceptance letter]

9 Dec 2024

PONE-D-24-28229R2 

PLOS ONE

Dear Dr. Kim, 

I'm pleased to inform you that your manuscript has been deemed suitable for publication in PLOS ONE. Congratulations! Your manuscript is now being handed over to our production team.

Kind regards, 

on behalf of

Dr. Charitha Dias 

Academic Editor

PLOS ONE